# SECOND: Mitigating Perceptual Hallucination in Vision-Language Models via Selective and Contrastive Decoding

**Woohyeon Park** [1]   **Woojin Kim** [1]   **Jaeik Kim** [2]   **Jaeyoung Do** [1 2]

## Abstract

Despite significant advancements in Vision-Language Models (VLMs), the performance of existing VLMs remains hindered by object hallucination, a critical challenge to achieving accurate visual understanding. To address this issue, we propose SECOND: Selective and Contrastive Decoding, a novel approach that enables VLMs to effectively leverage multi-scale visual information with an object-centric manner, closely aligning with human visual perception. SECOND progressively selects and integrates multi-scale visual information, facilitating a more precise interpretation of images. By contrasting these visual information iteratively, SECOND significantly reduces perceptual hallucinations and outperforms a wide range of benchmarks. Our theoretical analysis and experiments highlight the largely unexplored potential of multi-scale application in VLMs, showing that prioritizing and contrasting across scales outperforms existing methods. Code is available at https://github.com/AIDASLab/SECOND.

## 1. Introduction

Vision-Language Models (VLMs) have revolutionized Artificial Intelligence (Liu et al., 2024b;a) by effectively bridging vision and language modalities, enabling breakthroughs in tasks such as visual question answering (VQA) (Antol et al., 2015), image captioning (Vinyals et al., 2015), and open-vocabulary semantic segmentation (Zhang et al., 2023). These advancements have been driven by cross-modal alignment strategies and the integration of large-scale vision and language data. Despite these achievements, object halluci-

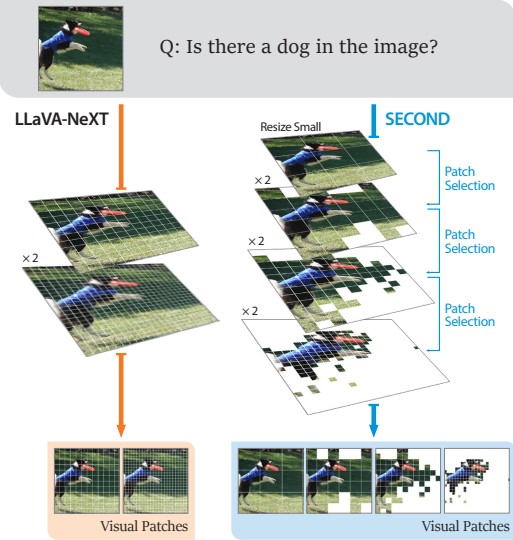

*Figure 1.* The comparison of LLaVA-NeXT (Liu et al., 2024a) and SECOND in utilizing visual patches. While LLaVA-NeXT uniformly incorporates patches at two scales, SECOND accumulates object-relevant patches through patch selection strategy.

nation—a phenomenon where models generate plausible yet inaccurate responses that are not grounded in the visual content—remains a persistent and critical challenge (Bai et al., 2024; Leng et al., 2024). This limitation poses a significant barrier to their adoption in high-stakes applications demanding reliability and interpretability.

Several studies have explored methods to mitigate hallucination in VLMs. These approaches include designing robust datasets to reduce model biases (Liu et al., 2023; Hu et al., 2023), suppressing the influence of LLM priors (Huang et al., 2024; Leng et al., 2024; Kim et al., 2024a), and improving the performance of vision models integrated into VLMs (Tong et al., 2024; Chen et al., 2024b). Moreover, recent VLMs, such as LLaVA-NeXT (Liu et al., 2024a), have incorporated multi-scale vision techniques (Li et al., 2024; Wang et al., 2024a), which have been effective in traditional computer vision tasks like object detection (Guan et al., 2022) and semantic segmentation (He et al., 2022b). However, prior research has largely focused on single-scale visual representations, overlooking the full potential of multi-scale integration.

[1]Department of Electrical and Computer Engineering, Seoul National University, Seoul, South Korea [2]Interdisciplinary Program in Artificial Intelligence, Seoul National University, Seoul, South Korea. Correspondence to: Jaeyoung Do <jaeyoung.do@snu.ac.kr>.

*Proceedings of the $42^{nd}$ International Conference on Machine Learning*, Vancouver, Canada. PMLR 267, 2025. Copyright 2025 by the author(s).

In this work, we align with the evolving trends in VLMs by optimizing multi-scale vision techniques to address object hallucination effectively. Our analysis reveals that current multi-scale VLM designs often integrate visual information indiscriminately across different scales, which can impede precise extraction of dense, object-specific details. This challenge arises due to the fundamentally spatial nature of visual data in contrast to textual data—where increasing the number of visual patches through up-scaling can lead to exponential information loss (He et al., 2022a).

To address these limitations, we propose SECOND (Selective and Contrastive Decoding), a novel, training-free framework designed to address object hallucination by leveraging multi-scale visual information with human-inspired selectivity and contrast (Purves et al., 2001). SECOND dynamically prioritizes object-relevant patches and refines visual representations through an iterative, multi-stage decoding process. By selectively incorporating coarse-to-fine-grained visual information and focusing on high-resolution relevant patches, SECOND enhances both visual fidelity and interpretability, effectively mitigating hallucination. As illustrated in Fig. 1, this process systematically filters out irrelevant patches, enabling the model to transition from broad-level scanning to fine-grained examination.

In addition, the multi-stage process of SECOND fosters a synergistic interaction between amateur and expert outputs, thereby introducing multi-stage Contrastive Decoding. The initial stage acts as an amateur, performing coarse image processing, while the final stage serves as an expert, delivering detailed and comprehensive analysis. This approach enhances step-by-step progression of SECOND by contrasting the expert, which utilizes fine-grained and relevant information from the image, with several amateurs across multiple stages. Unlike prior works on Contrastive Decoding in VLMs, which primarily focus on suppressing language priors (Leng et al., 2024; Kim et al., 2024a; Wang et al., 2024b), SECOND leverages selective contrast between sparse and dense visual information to effectively mitigate perceptual hallucinations. As a result, the expert output in the final stage of SECOND outperforms several baselines independently, and moreover, incorporating contrastive decoding further amplifies the improvement of performance (For more details, see Sec. 5).

- We propose SECOND, a novel, training-free framework that adaptively selects multi-scale patches and contrasts sparse visual data with fine-grained details.

- Our analysis of multi-scaled visual patches in VLMs offers insights into optimizing performance by balancing coarse and fine-grained visual information.

- Extensive experiments demonstrate that SECOND outperforms several baselines across diverse benchmarks, including POPE (Li et al., 2023), VQAv2 (Antol et al., 2015), MMStar (Chen et al., 2024a), and MM-Bench (Liu et al., 2025), highlighting its effectiveness.

## 2. Related Works

### 2.1. Vision-Language Model and Hallucination

The remarkable advancements in VLMs have been driven by the development of models such as LLaVA (Liu et al., 2024b), Instruct-BLIP (Dai et al., 2023), and Mini-GPT4 (Zhu et al., 2023), which have effectively bridged the modalities of vision and language. Despite these advancements, perceptual hallucination—a phenomenon in which VLMs generate compelling yet ungrounded object responses—remains a persistent challenge (Li et al., 2023). To mitigate this issue, various approaches have been explored. Training-based methods aim to enhance model robustness, designing robust dataset (Liu et al., 2023; Hu et al., 2023), employing contrastive loss (Jiang et al., 2024), or adopting Reinforcement Learning (RL) (Zhao et al., 2023; Sun et al., 2023). On the other hand, pre/post-processing methods are served in training-free manner. these works address hallucination leveraging auxiliary vision model (Tong et al., 2024; Chen et al., 2024b), refining visual input representations (Huang et al., 2024; Leng et al., 2024), or indiscriminately accumulating multi-scale visual features (Liu et al., 2024a; Li et al., 2024). Specifically, our approach identifies the limitation of the indiscriminate utilization of multi-scale vision, and introduces an optimal strategy for selectively integrating multi-scale fine-grained visual features without requiring additional training or auxiliary models.

### 2.2. Contrastive Decoding

Contrastive Decoding (CD) (Li et al., 2022), which contrasts the outputs of strong expert models with those of weak amateur models has emerged as a fundamental approach in LLMs, enhancing generation quality while ensuring factual consistency. Building on the success of contrastive decoding in LLMs, this framework has been extended to VLMs, with a primary focus on enhancing factual grounding in multi-modal generation tasks. VCD (Leng et al., 2024) utilizes noised images to distort visual inputs for contrastive output distributions, reducing biases and improving alignment with visual inputs. CODE (Kim et al., 2024a) employs self-generated visual descriptions as contrasting references to align textual outputs with visual content. While these approaches primarily focus on suppressing the prior knowledge of LLMs, HALC (Chen et al., 2024b) contrasts the specific cropped focal of image to enhance visual representation. Recent works further expand the single-step contrastive decoding paradigm through adaptive visual augmentations (Kim et al., 2024b; Park et al., 2025), diverse contrastive sampling and fusion strategies (Lee et al.,

2024), attention-guided ensembling (Cho et al., 2025), and token-level introspective filtering (Huo et al., 2025). Other approaches employ classifier-free grounding and prompt-based amplification (Wan et al., 2025; Favero et al., 2024b) to enhance multimodal alignment. In contrast to these methods, which rely on a single contrastive reference, we employ a multi-stage Contrastive Decoding, leveraging each intermediate output in SECOND as a contrastive reference, amplifying the distinction across multiple scales.

## 3. Core Principles

### 3.1. Problem Definition

VLMs, such as LLaVA (Liu et al., 2024b), tokenize visual information from image patches and integrate it with textual instructions. Given visual information $v$, an instruction $x$, and the model's generated answer $y$, we define the *hallucination probability* as the likelihood that $y$ diverges from the relevant information in $v$ (Favero et al., 2024a; Guerreiro et al., 2022).

**Definition 3.1.** *Hallucination probability. Given the probability $P(y \mid v, x)$ in VLMs, where $y$ denotes the generated text sequence for the given query $x$ and visual contents $v$, the probability of object hallucination is defined as $P_{Hal}$, representing the likelihood that the model does not generate the response $y$ based on the given query and visual content:*

$$P_{\text{Hal}} = 1 - P(y \mid v, x)$$
$$= 1 - \prod_{t=1}^{|L|} P(y_t \mid v, x, y_{i<t}),$$

where $t$ and $L$ each denote the timestep of the generation process and total length of generated tokens, with $y_t$ representing the token generated at timestep $t$ and $y_{i<t}$ representing all previously generated tokens.

Fig. 2(a) shows the distributions of $P_{\text{Hal}}$ for hallucinated and non-hallucinated answers by LLaVA on the POPE benchmark. Non-hallucinated responses predominantly exhibit lower hallucination probabilities, while hallucinated responses are more uniformly distributed. This distributional disparity highlights that reducing $P_{\text{Hal}}$ is essential for mitigating hallucination, aligning with Def. 3.1.

### 3.2. Theoretical Insights and Analysis

For VLMs to generate accurate responses based on visual content, we hypothesize that the model's ability to precisely focus on the target object plays a crucial role in determining the probability of hallucination (*i.e.*, $P_{\text{Hal}}$). To quantitatively analyze this, we measure the alignment between the model's attention and the object's segmentation mask. Specifically, we introduce the Attention Dice Coefficient

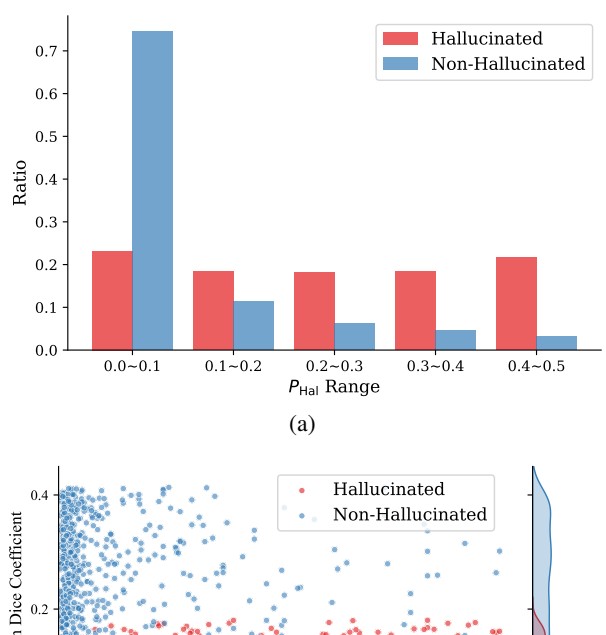

(a)

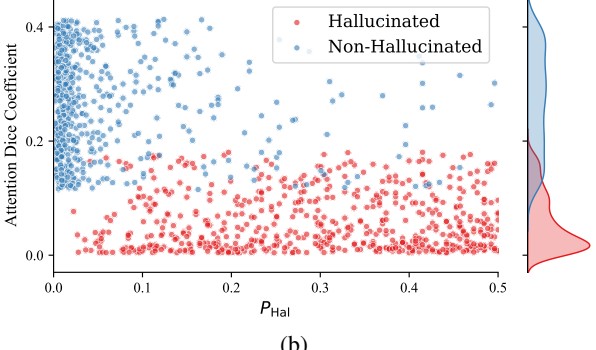

(b)

*Figure 2.* (a) Distribution of $P_{Hal}$ with the LLaVA model on POPE. (b) Attention Dice Coefficient on $P_{Hal}$

(Thm. 3.2), inspired by the widely used Dice Coefficient in image segmentation (Shamir et al., 2019), where a higher value indicates that the VLM is properly focusing on the relevant object.

Our approach integrates self-attention maps from the vision encoder and cross-attention maps between generated output tokens. To compute a unified attention signal that reflects both visual and textual modalities, we multiply these two attention maps element-wise. To assess this, we query the LLaVA model with *"Is there a/an {object} in the image?"* on MSCOCO (Lin et al., 2014), which provides mask annotations for each object. As shown in Fig. 2(b), we observe a strong correlation between the Attention Dice Coefficient and hallucination probability. Specifically, hallucinated responses (red points) tend to have lower Attention Dice scores and greater hallucination probabilities compared to non-hallucinated responses (blue points). These findings support our hypothesis that a model's ability to focus accurately on target objects significantly reduces hallucination risk, summarized in the following observation:

> ***Observation:*** $P_{\text{Hal}}$ decreases when model's visual attention aligns with the groundtruth object mask.

Based on this observation, we theoretically validate recent advancements in VLMs that aim to enhance visual information by utilizing small-sized patches (*i.e.*, visual scaling) with respect to their Attention Dice Coefficient scores as detailed in Thm. 3.2.

**Theorem 3.2.** *Consider an image composed of $n$ patches, each of size $m \times m$ pixels, where $\alpha_i \in [0, 1]$ represents the visual attention score of the $i$-th patch. The ground truth object mask assigns a value of 1 to pixels within the object region and 0 to the background. The average ground truth value for each patch is denoted as $g_i \in [0, 1]$. The* **Attention Dice Coefficient** *for an $m \times m$ pixel-sized patch is defined as follows:*

$$Dice_{attn}^{m} = \frac{2 \sum\limits_{i=1}^{n} \alpha_i g_i}{\sum\limits_{i=1}^{n} \alpha_i + \sum\limits_{i=1}^{n} g_i}.$$

*Given $m' < m$, it follows that $Dice_{attn}^{m} < Dice_{attn}^{m'}$,* **when the model's attention correctly aligns with the object mask**.

*Proof Sketch.* VLMs tend to show increased $Dice_{attn}$ in the absence of hallucinations (Fig. 2(b)), indicating higher attention $\alpha$ is assigned to patches, where containing higher value of $g$. Assume that each pixel in the ground truth mask follows a Bernoulli distribution with probability of $p \in [0, 1]$. Then, as the patch size $m$ increases, the mean of $g_i$ remains at $p$ while its standard deviation decreases as $\sqrt{p(1-p)}/m$. Consequently, the most of $g_i$ values concentrate around $p$, reducing variability and limiting the Attention Dice Coefficient's growth. In contrast, smaller patch sizes allow for more dynamic Attention Dice allocation, supporting recent VLM advancements that improve object focus and reduce hallucinations by up-scaling visual content.

However, simply reducing patch size to scale up the model has inherent limitations in improving performance (He et al., 2022a; Wang et al., 2024a; Beyer et al., 2023). This limitation is due to the indiscriminate scaling of all visual patches, which increases irrelevant information while diluting object-specific visual context. As the number of patches grows, we observe that VLMs struggle to precisely align their visual attention with ground-truth object masks. This misalignment induces perceptual hallucination by increasing the density of not only object-related but also background information. Instead of uniformly reducing patch size across the entire image, balancing relevance and quantity of visual content proves more effective in enhancing the Attention Dice Coefficient. To achieve this, we propose multi-stage patch selection, where object-relevant patches are iteratively selected at each scale and integrated. This approach preserves the density of object-relevant information while curbing the proliferation of patches unrelated to the object's focus.

In Thm. 3.3, we theoretically prove that multi-stage patch

selection increases the Attention Dice Coefficient, thereby reducing $P_{Hal}$.

**Theorem 3.3.** *Let $Dice_{attn}^{(s)}$ denote the Attention Dice coefficient at stage $s$ in a multi-stage framework. Multi-stage patch selection guarantees that the Attention Dice Coefficient at next stage, $s + 1$ is lower-bounded by the Dice coefficient at stage $s$, such that:*

$$Dice_{attn}^{(s)} \le Dice_{attn}^{(s+1)}$$

*Proof.* See Appendix A.

## 4. Methodology

Building on the theoretical foundations outlined in Sec. 3, we propose SECOND (Selective and Contrastive Decoding), a novel approach designed to mitigate perceptual hallucinations within a multi-stage framework. SECOND selectively integrates fine-grained information across stages based on the visual attention mechanisms of VLMs (Sec. 3.2) and enhances representation by contrasting enriched dense visual information with sparse representations at each stage.

### 4.1. Selective Multi-Scale Feature Integration

Since SECOND relies on effectively capturing multi-scale visual features, it requires an expansion of the vision encoder's scale. However, most existing models typically operate at only one or two scales (Liu et al., 2024b;a), and training additional multi-scale vision encoders for each VLM entails excessive computational costs. Therefore, a training-free approach for handling multiple resolutions is highly desirable. Inspired by prior studies (Dosovitskiy, 2020; Beyer et al., 2023), we employ bilinear interpolation on the positional embeddings of the vision encoder and down-sample input images to generate visual tokens at lower scales. By appending a vision encoder that processes down-sampled features to the existing model, we achieve multi-scale integration without additional training. Although the standalone performance of the down-sampled vision encoder is lower than the baseline, it remains effective for multi-scale integration, as demonstrated in Appendix B, proving its suitability for enhancing visual processing across multiple scales.

For a detailed implementation, we apply SECOND on recent VLMs as a baseline, leveraging their scale-aware trained capabilities to maximize performance in a training-free setting and effectively adapt to the emerging trend of visual scaling in VLMs. We choose the maximum length of stages as 4 to balance decoding computational efficiency and performance, which will be discussed in detail in Sec. 5.4. Given a visual encoder's base resolution before scaling up $H \times W$ with patch size $m \times m$, SECOND incorporates two additional visual encoders into the early stages: one processing a resolution of $H/4 \times W/4$ for the first stage

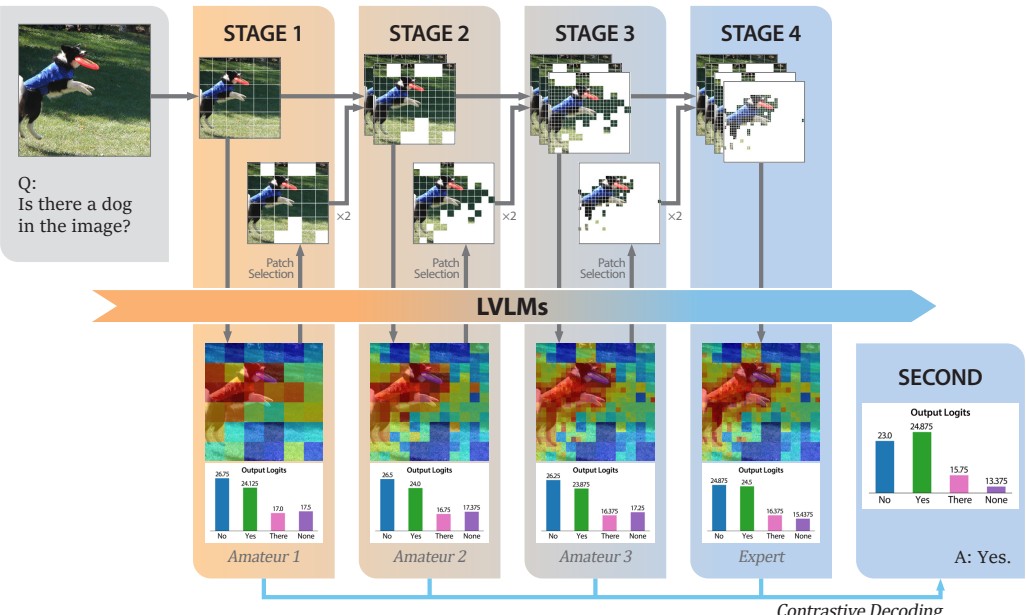

Figure 3. Architecture of SECOND. Starting with sparse visual information in Stage 1, SECOND incorporates higher scales' fine-grained information selectively based on visual attention. By contrasting each interval of stages, SECOND achieves the final output effectively mitigating perceptual hallucination. See Sec. 5.3 for more detailed view of visual attention.

and another processing $H/2 \times W/2$ for the second stage, both maintaining the same patch size. The vision encoder that processes the base resolution images of baseline VLMs constitutes the third stage in SECOND. For some models utilizing double-scale, such as LLaVA-NeXT, subsequently form the fourth stage, while single-scale models applied 3-stage design. This multi-stage approach preserves the scale gap between stages, effectively aligning with the existing twofold scaling utilized in VLMs. Furthermore, we adopt SECOND on VLMs utilizing CLIP (Radford et al., 2021) and SigLIP (Zhai et al., 2023), which are most widely used vision encoders. In the interpolation process, the embedding of the *cls* token in CLIP is preserved and does not participate in the interpolation, whereas SigLIP's positional embedding is fully interpolated due to the absence of a *cls* token. This modification allows the model to process low-resolution coarse images with fewer patches while gradually allocating more patches to high-resolution fine-grained images, thereby increasing visual density more robustly with the patch selection strategy described below.

As shown in Fig. 3, the process begins at Stage 1 by utilizing only the smallest scale vision encoder. At each subsequent stage, patches for larger-scale processing are selectively chosen based on the visual attention generated in the previous stage. As the stages progress, visual attentions at different scales are resized and accumulated to align with the attention map of the highest resolution. To selectively filter out object relevant patches, $p_{select}$, we introduce a dynamic

patch selection mechanism that combines a hyperparameter $\lambda$, with the entropy of the visual attention $H(V)$. Low entropy in the visual attention indicates a clear distinction between the attention on objects and the background, suggesting that fewer additional patches are required for the next stage. In contrast, high entropy indicates that the model fails to focus appropriately on specific objects, necessitating the incorporation of more fine-grained information in the subsequent stage. Therefore, the patch selection percentage $p_{select}$ is designed as below:

$$p_{select} = \frac{exp(\lambda * H(V)) - 1}{exp(\lambda) - 1}.$$

By selecting $p_{select}\%$ patches with highest visual attention in each stage, SECOND intensively accumulates coarse-to-fine-grained visual information through multiple stages. A detailed algorithmic description of the patch selection process is provided in Appendix D.

### 4.2. Multi-Stage Contrastive Decoding

Contrastive Decoding (CD) (Li et al., 2022) builds upon the perspective that when a powerful expert model fails to perform optimally, a weaker amateur model is likely to encounter even more pronounced failures. By contrasting the output logits of these two models with hyperparameter $\alpha$, single-stage CD amplifies the output quality of expert model as following:

$$logit_{Single} = logit_{expert} + \alpha(logit_{expert} - logit_{amateur}).$$

*Table 1.* Results of POPE in SECOND and baselines. SECOND outperformed in 11 out of 12 highlighted cases. Inside the parentheses below each model name indicates the type of vision model and its base resolution.

| Model | LLM | Method | CD | MSCOCO | | | OKVQA | | | GQA | | |
|---|---|---|---|---|---|---|---|---|---|---|---|---|
| | | | | Recall (↑) | Acc. (↑) | f1 (↑) | Recall (↑) | Acc. (↑) | f1 (↑) | Recall (↑) | Acc. (↑) | f1 (↑) |
| LLaVA-Next (CLIP-336) | Vicuna-7B | baseline | ✗ | 78.8 | 87.7 | 86.5 | 86.7 | 89.1 | 88.8 | 84.8 | 86.6 | 86.3 |
| | | VCD | ✓ | 81.1 | 88.2 | 87.3 | 88.3 | 88.0 | 88.1 | 86.3 | 84.6 | 84.9 |
| | | SECOND | ✗ | 80.1 | 88.6 | 87.5 | 87.6 | 89.9 | 89.6 | 84.9 | 86.5 | 86.3 |
| | | SECOND | ✓ | 85.1 | 89.7 | **89.2** | 90.5 | 90.3 | **90.4** | 85.5 | 89.4 | **87.4** |
| | Mistral-7B | baseline | ✗ | 80.2 | 88.3 | 87.3 | 88.2 | 88.7 | 88.7 | 88.2 | 84.2 | 84.8 |
| | | VCD | ✓ | 80.8 | 87.4 | 86.6 | 88.0 | 88.2 | 88.3 | 88.0 | 84.5 | 85.1 |
| | | SECOND | ✗ | 79.5 | 88.1 | 86.9 | 86.8 | 88.4 | 88.2 | 87.2 | 84.8 | 85.2 |
| | | SECOND | ✓ | 84.8 | 89.3 | **88.8** | 92.5 | 89.9 | **90.7** | 92.1 | 85.3 | **87.5** |
| LLaVA-OneVision (SigLIP-384) | Qwen2-0.5B | baseline | ✗ | 80.0 | 88.4 | **87.4** | 85.6 | 89.3 | 88.9 | 83.1 | 86.8 | 86.3 |
| | | VCD | ✓ | 79.7 | 87.4 | 86.4 | 86.7 | 89.0 | 88.7 | 84.3 | 86.7 | 86.4 |
| | | SECOND | ✗ | 78.1 | 87.6 | 86.3 | 83.8 | 88.7 | 88.1 | 82.2 | 87.4 | 86.7 |
| | | SECOND | ✓ | 79.7 | 87.9 | 86.9 | 85.4 | 89.3 | **89.1** | 83.3 | 87.8 | **87.2** |
| Yi-VL (CLIP-448) | Yi-6B | baseline | ✗ | 70.3 | 82.0 | 79.6 | 77.0 | 84.0 | 82.8 | 74.5 | 81.0 | 79.7 |
| | | VCD | ✓ | 73.0 | 80.1 | 78.6 | 79.1 | 82.1 | 81.6 | 78.2 | 79.9 | 79.5 |
| | | SECOND | ✗ | 73.5 | 83.5 | 82.0 | 80.1 | 85.6 | 84.8 | 76.6 | 82.5 | 81.4 |
| | | SECOND | ✓ | 83.4 | 84.5 | **84.3** | 87.7 | 86.3 | **86.5** | 83.3 | 82.8 | **82.9** |

Obviously, typical CD can be applied to SECOND, utilizing the initial stage output as an amateur and the final one as an expert. However, our focus is on the multi-stage design of SECOND, which generates hierarchically structured outputs that progressively highlight the differences across stages.

Through patch selection across multiple stages, SECOND progressively integrates fine-grained visual information, resulting in increasingly refined outputs. The outputs generated at each stage reflect the evolutionary progression from the initial to the final stage (Thm. 3.3), forming a performance-ordered hierarchy. Leveraging these prioritized multiple outputs, SECOND facilitates the application of multi-staged CD. This framework effectively amplifies the difference of visual information driven by scale interval and patch selection rather than simply contrasting the most amateur and expert (Sec. 5.4). To utilize the evolutionary trajectory of stage-wise outputs, we propose a multi-staged CD framework incorporating contrasting hyperparameters $\alpha$, $\beta$, and $\gamma$ defined within the range of 0 to 1 as follows:

$$\text{logit}_{\text{SECOND}} =$$
$$\text{logit}_{\text{expert}} + \alpha(\text{logit}_{\text{expert}} - \text{logit}_{\text{amateur3}})$$
$$+ \beta(\text{logit}_{\text{amateur3}} - \text{logit}_{\text{amateur2}})$$
$$+ \gamma(\text{logit}_{\text{amateur2}} - \text{logit}_{\text{amateur1}}).$$

By applying this Multi-staged CD, SECOND amplifies the effectiveness of patch selection at each stage. This enables the comparison between the inferences derived from sparse visual information in the early stages and those obtained from progressively incorporated dense and fine-grained visual information. As a result, this approach facilitates significant performance improvements in object perception tasks.

## 5. Experiments

To evaluate SECOND, we implement SECOND on three recent VLMs. First, LLaVA-NeXT (Liu et al., 2024a), an enhanced version of LLaVA, incorporates one additional scaled-up features using CLIP as a vision model with resolution of 336. We evaluate this model with both the Vicuna-7B and Mistral-7B LLMs. LLaVA-Onevision (Li et al., 2024), similar to LLaVA-NeXT in applying a single level of scale-up but differs in employing SigLIP-384 as the vision encoder instead of CLIP-336. To validate the efficacy of SECOND on smaller LLMs, we conduct experiments using Qwen2-0.5B. Unlike the previous two models, Yi-VL (Young et al., 2024) employs single-scale image features. We apply SECOND to this single-scale model to show its effectiveness even in models trained with single-scale features. Furthermore, we incorporate VCD (Leng et al., 2024), a Contrastive Decoding framework that utilizes noised image, enabling comparative analysis on other decoding method. For the several hyperparameters in SECOND, we serve the optimal settings in Appendix C, further analyzing the hyperparameter sensitivity in Sec. 5.5.

### 5.1. Benchmarks and Evaluation Metricss

**POPE** POPE (Li et al., 2023) is a widely adopted benchmark that specializes in identifying perceptual hallucination by querying the presence of specific objects in a given image through simple *yes/no* questions. It employs recall, accuracy, and f1 score as the primary evaluation metrics and includes 3k questions derived from well-known datasets such as MSCOCO (Lin et al., 2014), A-OKVQA (Schwenk et al., 2022), and GQA (Hudson & Manning, 2019). In this study, we evaluated the models using the popular split of the POPE benchmark.

*Table 2.* Result of various benchmarks including VQAv2(lite), MMStar, and MMBench(lite). Inside the parentheses below each model name indicates the type of vision model and its resolution.

| Model | LLM | Method | CD | VQAv2 (lite) | MMStar | MMbench (lite) |
|---|---|---|---|---|---|---|
| LLaVA-Next (CLIP-336) | Vicuna-7B | baseline | ✗ | 76.4 | 37.3 | 75.8 |
| | | VCD | ✓ | 72.9 | 38.1 | 74.2 |
| | | SECOND | ✗ | 76.5 | 37.5 | 78.0 |
| | | SECOND | ✓ | **77.5** | **38.6** | **80.0** |
| | Mistral-7B | baseline | ✗ | 72.0 | 32.4 | **74.2** |
| | | VCD | ✓ | 70.1 | 34.0 | 71.2 |
| | | SECOND | ✗ | 73.6 | 34.1 | 71.2 |
| | | SECOND | ✓ | **74.5** | **36.2** | 70.5 |
| LLaVA-OneVision (SigLIP-384) | Qwen2-0.5B | baseline | ✗ | 74.6 | 38.9 | **73.5** |
| | | VCD | ✓ | 55.0 | 36.2 | 70.5 |
| | | SECOND | ✗ | 73.6 | 39.6 | 72.7 |
| | | SECOND | ✓ | **75.1** | **39.9** | **73.5** |
| Yi-VL (CLIP-448) | Yi-6B | baseline | ✗ | 64.3 | 34.8 | 77.3 |
| | | VCD | ✓ | 61.9 | 34.4 | 79.5 |
| | | SECOND | ✗ | 63.6 | 37.4 | 82.6 |
| | | SECOND | ✓ | **65.3** | **39.8** | **84.8** |

**General Tasks** For the general tasks, VQAv2 (Antol et al., 2015) serves as a benchmark for evaluating VLMs' ability to generate answers for given image-question pairs. VQAv2 measures performance through an exact match evaluation, requiring models to provide answers in the form of *yes/no*, numerical, or open-ended responses. We evaluate the lite version consisting of 0.5k questions, which encompass a wide range of categories, including object presence, attributes, and relationships. MMStar and MMBench (Chen et al., 2024a; Liu et al., 2025) provide comprehensive evaluations by categorizing VLM tasks into two primary domains: perception and reasoning. These benchmarks further divide main categories into several sub-categories, enabling a detailed assessment of model performance across a variety of tasks. MMStar comprises 1.5k questions, while MMBench's lite version includes 0.5k samples. These benchmarks collectively offer a holistic view of VLMs' capabilities, ensuring a balanced evaluation across both low-level perception and high-level reasoning tasks.

## 5.2. Main results

**Results on POPE** As presented in Tab. 1, across the experimental configurations of POPE, SECOND outperformed the baseline and VCD method in 11 out of 12 cases. These results provide compelling evidence that SECOND effectively mitigates perceptual hallucinations while substantially enhancing the object recognition capabilities of VLMs.

**Results on General Tasks** In addition to general tasks, as illustrated in Tab. 2, SECOND outperformed most baselines. The enhanced performance observed in VQAv2, which involves object perceptual queries in an open-ended format, indicates that SECOND is effective not only in single-word closed-answer tasks like POPE but also in handling open-

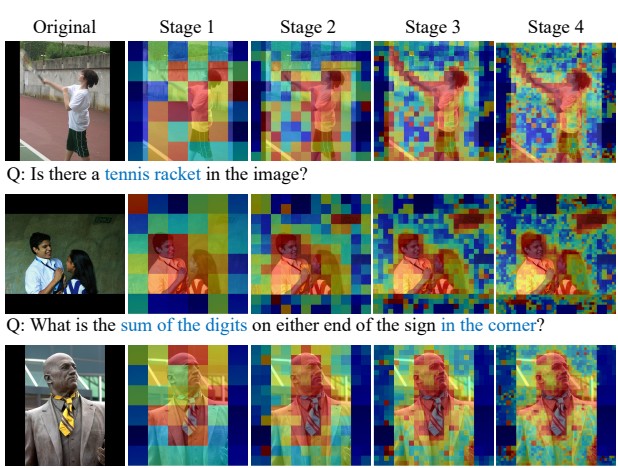

Q: Is there a tennis racket in the image?

Q: What is the sum of the digits on either end of the sign in the corner?

Q: What part of the statue's outfit isn't made of the same material as the statue?

*Figure 4.* Evolutionary attention refinement in SECOND. Red areas represent high attention scores. SECOND gradually enhances visual attention, demonstrating robustness across diverse object scales.

ended questions. Furthermore, the performance improvements observed in MMStar and MMBench, which include both perception and reasoning tasks, highlight that SECOND's approach does not negatively impact the reasoning capabilities of LLMs. These results also suggest that improvements in perception performance can translate into benefits for higher-level tasks such as reasoning, underscoring the holistic advantages of SECOND.

## 5.3. Robustness of SECOND

Leveraging multi-scale visual information that has not been seen during training, and executing multiple stages in a sequential manner, requires that the algorithm operates robustly. Therfore, we demonstrate the robustness of SECOND

*Table 3.* Effect of individual stages on POPE MSCOCO benchmark performance and decoding time with LLaVA-NeXT

| Stages | POPE (f1) | | Decoding Time |
|---|---|---|---|
| | w.o. CD | w. CD | |
| 42-84-168-336-672 | 87.6 | 89.3 | 1.4x |
| 84-168-336-672 | 87.5 | 89.2 | 1x |
| 168-336-672 | 87.5 | 88.4 | 0.7x |
| 336-672 | 87.2 | 87.7 | 0.6x |

*Table 4.* Effect of diverse patch selection strategy on POPE MSCOCO benchmark performance with LLaVA-NeXT

| Stages | Patch Selection | POPE (f1) |
|---|---|---|
| 84-168-336-672 | dynamic | 87.5 |
| 84-168-336-672 | reversed | 70.6 |
| 84-168-336-672 | fixed (30%) | 78.0 |
| 84-168-336-672 | fixed (50%) | 83.8 |
| 84-168-336-672 | fixed (70%) | 86.1 |
| 42-84-168-336-672 | all (100%) | 87.2 |
| 84-168-336-672 | all (100%) | 87.1 |
| 168-336-672 | all (100%) | 87.1 |

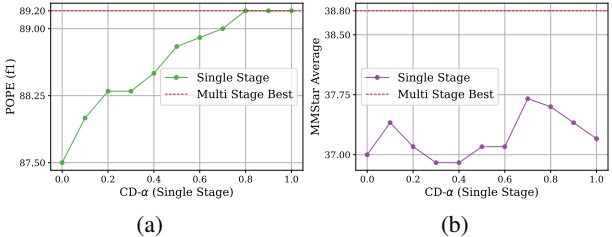

(a)          (b)

*Figure 5.* (a) Results on POPE across CD-$\alpha$. (b) Results on MM-Star, comparing single CD to the best of multi-stage CD.

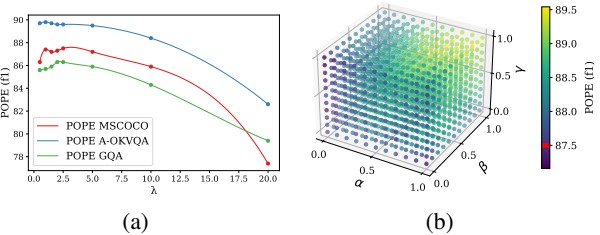

(a)          (b)

*Figure 6.* Hyperparameter sensitivity analysis. (a) $\lambda$ in patch selection. (b) $\alpha, \beta$ and $\gamma$ in multi-stage CD. The red line in (b) represents the performance of SECOND w.o. CD.

in Fig. 4. In the first example, although the side of the tennis racket is captured and thus poorly visible, the attention map is eventually corrected as fine-grained information is added in later stages. In the second example, the small text on the upper-right wall initially receives no attention from the model. However, as the stages progress, the relevant visual cues are gradually emphasized. Because relevant area is not attended on the first stage, Contrastive Decoding can even amplify the probability of generating the correct answer. In the final example, the model initially over-focuses on the shoulder area of the statue in the first stage. As the stages proceed, this unnecessarily strong attention weakens, demonstrating refinement of focus.

### 5.4. Ablation study

**Impact of Stage Configuration** In our multi-stage generation process, we conduct ablation studies to systematically analyze the influence of stage configuration, justifying the selection of our default setting. Tab. 3 provides a detailed analysis of how different stage configurations impact POPE benchmark performance and decoding time with LLaVA-NeXT. Each column under *Stages* corresponds to the specific multi-scale resolutions utilized in the generation process. The results show that the performance without contrastive decoding (w.o. cd) is relatively stable across configurations, though a notable performance gap exists between the 84-168-336-672 and 168-336-672 settings when contrastive decoding is applied. While the inclusion of the 42 resolution slightly improves performance (0.1 in POPE), the effect is not significant enough to justify the increased

decoding time complexity (1.4x compared to 84). Therefore, the 84-168-336-672 configuration is selected as the baseline, which balances performance and decoding efficiency.

**Selection Strategy Analysis** To validate our patch selection mechanism and its theoretical basis for reducing hallucinations, we perform an ablation study summarized in Tab. 4. The default dynamic selection strategy across four stages achieves the highest performance (87.5), demonstrating its adaptability in selecting salient regions. Using all patches at varying stages consistently results in lower scores, highlighting the critical importance of selective patch utilization in optimizing performance and minimizing hallucinations. Fixed selection strategies (e.g. 30%, 50%, and 70%) further underscore this point, with the 70% fixed strategy achieving 86.1 but still falling short of the dynamic approach.

**Single vs. Multi Stage Contrastive Decoding** To evaluate the advantages of multi-stage CD over single-stage, we compare their performance under varying configurations of the CD-$\alpha$ parameter, which is adjusted between 0 and 1. In the single-stage approach, stage 4 logits are used as the primary outputs, while stage 1 logits serve as the contrastive reference, following the adapted procedure described in Sec. 4.2. Fig. 5 shows that single-stage CD achieves comparable performance to multi-stage CD in POPE, reaching the same peak value of 89.2 with optimal CD-$\alpha$. However, for the more general mmstar metric, multi-stage CD outperforms single-stage CD, achieving a higher peak value of 38.8, demonstrating its robustness in broader scenarios.

## 5.5. Hyperparameter Sensitivity

**Patch Selection parameter** In the patch selection process, we introduce the hyperparameter $\lambda$, which controls the number of regions selected within the visual attention as incremental patches for the next stage. As the value of $\lambda$ increases, fewer regions are selected. Fig. 6(a) illustrates the sensitivity of SECOND to varying $\lambda$ values. This analysis, combined with the preceding evaluation of selection strategies, highlights the importance of selecting patches at an appropriate ratio to achieve optimal performance. Practically, we observe that $\lambda = 1.0$ achieves consistently high performance across a tasks (Tab. 6 in Appendix C); therefore, we recommend using $\lambda = 1.0$ as the default when deploying SECOND.

**Multi-stage Contrastive Decoding parameters** In the multi-stage Contrastive Decoding framework of SECOND, the number of hyperparameters scales with the number of stages. In the default 4-stage configuration of SECOND, three key hyperparameters, $\alpha$, $\beta$, and $\gamma$, are used. As shown in Fig. 6(b), these parameters are explored by varying their values between 0 and 1 in increments of 0.1, resulting in 1,331 configurations. All configurations consistently show superior performance compared to the baseline LLaVA-NeXT, with 98.7% surpassing the non-Contrastive Decoding, which relies solely on patch selection.

## 6. Conclusion

In this paper, we propose SECOND (Selective and Contrastive Decoding), a novel framework that leverages multi-scale visual patch selection in Vision-Language Models (VLMs) and contrasts visual information across varying densities. Based on our analysis of perceptual hallucinations, SECOND demonstrates remarkable improvements in object-focused experiments while effectively mitigating hallucinations. These advancements enhance the foundational capabilities of VLMs for higher-level tasks and provide valuable insights into the application of multi-scale approaches in VLMs. Despite these promising results, SECOND's dependence on internal attention maps can fail when attention is poorly calibrated. Future work should therefore concentrate on strengthening the core selection mechanism to ensure relevant patches are reliably identified.

## Acknowledgements

This work was supported in part by National Research Foundation of Korea (NRF) grant (RS-2025-00560762, RS-2024-00414981), Institute of Information & communications Technology Planning & Evaluation (IITP) grant (IITP-2025-RS-2024-00397085, RS2021-II211343), and Seoul National University grant (Creative-Pioneering Researchers Program, Research Grant 0418-20240053). J. Do is with ASRI, Seoul National University.

## Impact Statement

This paper presents work whose goal is to advance the field of Machine Learning. There are many potential societal consequences of our work, none which we feel must be specifically highlighted here.

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

# A. Proof of Theorem 3.3

**Theorem 3.3** Let $\text{Dice}_{\text{attn}}^{(s)}$ denote the Attention Dice coefficient at stage $s$ in a multi-stage framework. Multi-stage patch selection guarantees that the Attention Dice Coefficient at next stage, $s + 1$ is lower-bounded by the Dice coefficient at stage $s$, such that:

$$\text{Dice}_{\text{attn}}^{(s)} \leq \text{Dice}_{\text{attn}}^{(s+1)}.$$

The updated attention score $\alpha^{(s+1)}$ by selectively incremented patches from stage $s$ to stage $(s + 1)$ is given by:

$$\alpha^{(s+1)} = \alpha^{(s)} + \Delta\alpha.$$

Thus,

$$\text{Dice}_{\text{attn}}^{(s+1)} = \frac{2\sum\limits_{i=1}^{n}(\alpha_i^{(s)} + \Delta\alpha_i)g_i}{\sum\limits_{i=1}^{n}(\alpha_i^{(s)} + \Delta\alpha_i) + \sum\limits_{i=1}^{n}g_i}.$$

Replace number of patches $n$ with number of pixels $N$. Then,

$$\text{Dice}_{\text{attn}}^{(s+1)} = \frac{2\sum\limits_{j=1}^{N}(\alpha_j^{(s)} + \Delta\alpha_j)g_j}{\sum\limits_{j=1}^{N}(\alpha_j^{(s)} + \Delta\alpha_j) + \sum\limits_{j=1}^{N}g_j}.$$

**Claim:** Attention Dice coefficient at next stage, $s + 1$ is lower-bounded by the Dice coefficient at stage $s$.

$$\text{Dice}_{\text{attn}}^{(s)} \leq \text{Dice}_{\text{attn}}^{(s+1)} \Leftrightarrow$$

$$\frac{2\sum\limits_{j=1}^{N}\alpha_j^{(s)}g_j}{\sum\limits_{j=1}^{N}\alpha_j^{(s)} + \sum\limits_{j=1}^{N}g_j} \leq \frac{2\sum\limits_{j=1}^{N}(\alpha_j^{(s)} + \Delta\alpha_j)g_j}{\sum\limits_{j=1}^{N}(\alpha_j^{(s)} + \Delta\alpha_j) + \sum\limits_{j=1}^{N}g_j}.$$

Split domain within $mask$ or not($\sim mask$). Then,

$$\frac{2\sum\limits_{mask}\alpha_j^{(s)}g_j}{\sum\limits_{mask}\alpha_j^{(s)} + \sum\limits_{\sim mask}\alpha_j^{(s)} + \sum\limits_{mask}g_j} \leq \frac{2\sum\limits_{mask}(\alpha_j^{(s)} + \Delta\alpha_j)g_j}{\sum\limits_{mask}(\alpha_j^{(s)} + \Delta\alpha_j) + \sum\limits_{\sim mask}(\alpha_j^{(s)} + \Delta\alpha_j) + \sum\limits_{mask}g_j}.$$

Supposing selected patches are only on the mask with $g = 1$,

$$\frac{2\sum\limits_{mask}\alpha_j^{(s)}}{\sum\limits_{mask}\alpha_j^{(s)} + \sum\limits_{\sim mask}\alpha_j^{(s)} + area(mask)} \leq \frac{2\sum\limits_{mask}(\alpha_j^{(s)} + \Delta\alpha_j)}{\sum\limits_{mask}(\alpha_j^{(s)} + \Delta\alpha_j) + \sum\limits_{\sim mask}(\alpha_j^{(s)}) + area(mask)} \Leftrightarrow$$

$$0 \leq \sum\limits_{mask}\Delta\alpha^{(s)} \times (\sum\limits_{\sim mask}(\alpha^{(s)} + area(mask)).$$

## B. Experiment on down-sampled Vision Encoders

This section details the performance of each stage when used independently. We evaluate each stage with LLaVA-NeXT on POPE MSCOCO popular split and MMStar in Tab. 5.

*Table 5.* Performance of down-sampled vision encoders in stand-alone setting

| Stages | POPE (Acc.) | MMStar |
|--------|-------------|--------|
| 84     | 51.2        | 25.9   |
| 168    | 74.4        | 34.5   |
| 336    | 84.9        | 37.4   |
| 672    | 88.3        | 34.0   |

## C. Optimal Hyperparameter Settings in SECOND

This section details in the optimal settings of hyperparamerts in SECOND. Tab. 6 details in the optimal settings of patch selection hyperparameter $\lambda$, and Tab. 7 details in the multi-stage Contrastive Decoding hyperparameters, $\alpha$, $\beta$, and $\gamma$. In Yi-VL, $\gamma$ is not utilized as the model is inherently designed for a single-stage process and does not include Stage 4.

*Table 6.* Optimal settings of patch selection hyperparameter $\lambda$.

| Model | LLM | POPE | VQAv2 | MMStar | MMBench |
|-------|-----|------|-------|--------|---------|
| LLaVA-NeXT | Vicuna-7B | 2.5 | 0.5 | 7.0 | 2.0 |
|            | Mistral-7B | 1.0 | 3.0 | 1.0 | 1.0 |
| LLaVA-OneVision | Qwen2-0.5B | 1.0 | 1.0 | 1.0 | 1.5 |
| Yi-VL | Yi-6B | 2.0 | 4.0 | 1.0 | 3.0 |

*Table 7.* Optimal settings of multi-stage CD hyperparameters $\alpha$, $\beta$, and $\gamma$.

| Model | LLM | parameter | POPE | | | VQAv2 | MMStar | MMBench |
|-------|-----|-----------|--------|---------|-----|-------|--------|---------|
|       |     |           | MSCOCO | A-OKVQA | GQA |       |        |         |
| LLaVA-NeXT | Vicuna-7B | $\alpha$ | 0.7 | 0.3 | 1.0 | 0.1 | 1.0 | 0.4 |
|            |           | $\beta$  | 0.7 | 0.7 | 0.7 | 0.2 | 0.8 | 0.2 |
|            |           | $\gamma$ | 1.0 | 0.4 | 1.0 | 0.1 | 0.4 | 0.6 |
|            | Mistral-7B | $\alpha$ | 0.5 | 0.9 | 0.9 | 0.1 | 0.1 | 0.1 |
|            |            | $\beta$  | 0.5 | 1.0 | 1.0 | 0.3 | 0.3 | 0.4 |
|            |            | $\gamma$ | 0.4 | 0.4 | 0.6 | 0.1 | 0.4 | 0.4 |
| LLaVA-OneVision | Qwen2-0.5B | $\alpha$ | 0.4 | 0.5 | 0.5 | 0.2 | 0.1 | 0.1 |
|                 |            | $\beta$  | 0.6 | 1.0 | 0.8 | 0.3 | 0.2 | 0.4 |
|                 |            | $\gamma$ | 0.8 | 0.7 | 0.6 | 0.4 | 0.1 | 0.1 |
| Yi-VL | Yi-6B | $\alpha$ | 0.8 | 0.6 | 0.1 | 0.4 | 1.0 | 0.0 |
|       |       | $\beta$  | 0.9 | 0.7 | 1.0 | 0.2 | 0.7 | 0.6 |

# D. Patch Selection Algorithm

---

**Algorithm 1** Patch Selection Algorithm

---

**Require:**     Stages list *stages*, image feature $v$, image masks $masks$, instruction $x$,
    patch selection hyperparameter $\lambda$,
**output**     Model response $y$.

  1:                                                **// Initialize image_masks for each stage**
  2: **for** *stage* in *stages* **do**
  3:    **if** *stage* = Stage 1 **then**
  4:       $masks[stage] \leftarrow$ ones()                         // All-ones tensor for the first stage
  5:    **else**
  6:       $masks[stage] \leftarrow$ zeros()                   // All-zeros tensor for the subsequent stages
  7:    **end if**
  8: **end for**
  9:                                 **// Process each stage, compute attentions, and update masks**
10: **for** *stage* in *stages* **do**
11:    $y \leftarrow$ model$(v \times masks[stage], x)$
12:    **if** *stage* = Final Stage **then**
13:       **break**                                 // Stop if this is the last stage
14:    **end if**
15:
16:    $SelfAttn \leftarrow$ Get *vision model's self-attention*
17:    $CrossAttn \leftarrow$ Get *language model's cross-attention*
18:    $VisualAttn \leftarrow$ zeros()                // Initialize *VisualAttn* to All-zeros tensor
19:    **for** $weight$ in $CrossAttn$ **do**
20:       $VisualAttn \leftarrow VisualAttn + (weight \times SA)$
21:    **end for**
22:
23:    $H \leftarrow$ get_entropy$(VisualAttn)$
24:    $p_{select} \leftarrow \dfrac{\exp(\lambda \times H) - 1}{\exp(\lambda) - 1}$       // Calculate patch selection percentage $p_{select}$ in Sec. 4.1
25:    $threshold \leftarrow$ topk$(VisualAttn, \text{len}(VisualAttn) \times p_{select})$.values$[-1]$
26:    $masks[stage + 1] \leftarrow (VisualAttn > threshold)$
27: **end for**

---

# E. Hallucination probabilty on each stage

Fig. 7 (a) and (b) illustrates percentage of $P_{Hal}$ on each stage with or without hallucination.

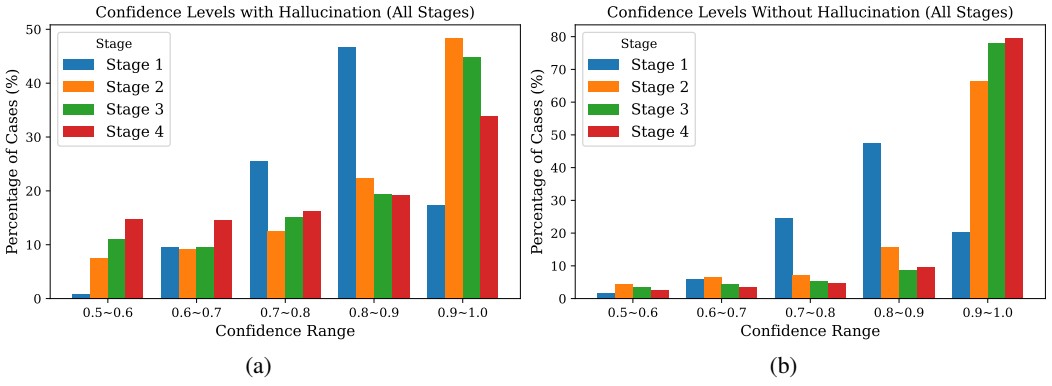

(a)                             (b)

*Figure 7.* (a) Percentage of $P_{Hal}$ on each stage with hallucination. (b) Percentage of $P_{Hal}$ without hallucination.

