# OpenReview forum: "SECOND: Mitigating Perceptual Hallucination in Vision-Language Models via Selective and Contrastive Decoding"
_ICML.cc/2025/Conference — ICML 2025 poster_

### Official Review · Reviewer_i6dn · 2025-03-10

**Overall Recommendation:** 3

**Summary:**

The paper addresses the problem of visual hallucinations in LVLMs by introducing a training-free framework called SECOND, which adaptively selects visual patches on multiple scales and applies Contrastive Decoding (CD) between the intermediate stage logits and the logits from the fine-grained expert. The paper provides theoretical motivation for the method and an empirical analysis on the hallucinations using the established benchmarks.

**Claims And Evidence:**

The authors propose the SECOND method, which is a training-free framework to be integrated to VLMs to dynamically select more precise visual information from multi-scale patches to reduce the object hallucinations in VLMs. The authors provide an initial analysis on hallucinations which motivates the method, and evaluate SECOND on meaningful benchmarks. However, the authors could better show that SECOND is able to correctly select either broad or fine-grained visual information depending on the task.

**Essential References Not Discussed:**

As discussed above, the following relevant works are not mentioned nor compared to:

- Paying more attention to image: A training-free method for alleviating hallucination in lvlms, ECCV 2024

- Contrastive Region Guidance: Improving Grounding in Vision-Language Models without Training, ECCV 2024.

- OPERA: Alleviating Hallucination in Multi-Modal Large Language Models via Over-Trust Penalty and Retrospection-Allocation, CVPR 2024

**Experimental Designs Or Analyses:**

- How were  \alpha, \beta, \gamma, and \lambda determined in the experiments? Table 6 and 7 show that these parameters are not robust to changes in the data and models. If I want to use your method, how do I know how to set these hyperparameters?

- While experiments do show benefits of SECOND across different setups, there are still some cases where performance gains are marginal or even negative which makes me doubtful about its usefulness in practice. For example, in Table 1, baseline outperforms both VCD and SECOND on MSCOCO in row 3. Similar results are seen for Mistral-7B in Table 2. There is no discussion in the paper about this, despite the fact that computational cost is significantly increased. Is this a consequence of poor hyperparameter choice, specifics of the model or the benchmark? Please elaborate on these results.

- Related to above, how does computational complexity of SECOND compare to that of VCD?

**Methods And Evaluation Criteria:**

While the chosen benchmarks do make sense, the proposed SECOND method is compared to only one baseline method but none of the following:

- Paying more attention to image: A training-free method for alleviating hallucination in lvlms, ECCV 2024

- Contrastive Region Guidance: Improving Grounding in Vision-Language Models without Training, ECCV 2024.

- OPERA: Alleviating Hallucination in Multi-Modal Large Language Models via Over-Trust Penalty and Retrospection-Allocation, CVPR 2024

**Other Comments Or Suggestions:**

- Presentation of the method could be improved with more details about the method.

- nit: L not defined in def 3.1

**Other Strengths And Weaknesses:**

See other comments

**Questions For Authors:**

See the comments above

**Relation To Broader Scientific Literature:**

Reducing object hallucinations in VLMs is an important topic and the paper introduces a novel approach combining CD and multiscale patch selection. This differentiates it from other proposed works which are typically proposing to contrast the output distributions of multimodal inputs with those of text-only inputs.

**Theoretical Claims:**

In my view, the current version of the paper doesn't sufficiently explain the details of the proposed method:

- it is unclear to me how attention is translated or mapped across different resolutions,

- it is not clear how transitions between the stages are performed. In my understanding, at the end of the first stage we end up with a subset of the selected patches. In the second stage, we start with an image of higher resolution but I do not understand how the patches from the first stage are used here? Is the image always sliced to the same number of patches at each stage?

---

> ### Author Rebuttal · Authors · 2025-04-01
>
> Dear Reviewer i6dn,
>
> We greatly appreciate your valuable feedback on our paper. We address the raised concerns and questions below.
>
> **W1: Essential References Not Discussed**
>
> Thank you for the suggestion. We have added the mentioned works to Sec. 2. In particular, we included OPERA in our computational cost comparison (see KN5t W2).
>
> **W2: It is unclear how attention is translated or mapped across different resolutions.**
>
> As described in Sec. 4.1, we extract attention maps at multiple resolutions using different patch sizes. These are resized to a common resolution and summed to form a unified attention map that captures multi-scale cues.
>
> **W3: It is not clear how transitions between the stages are performed.**
>
> We appreciate the feedback and would like to clarify. In SECOND, patch selection is not applied at the initial stage. The model first performs inference using tokens covering the full image, generating an attention map. Based on its attention entropy, the patch selection ratio for the next stage is determined.
> In the second stage, high-resolution patches from selected regions are added for inference. This aligns with multi-scale strategies like LLaVA-OneVision, where features at different resolutions are combined. The updated attention map reflects both the initial full-image attention and the influence of newly added high-res patches.
> Stage transitions are dynamic—SECOND may add patches in regions that gain attention over time. As shown in Fig. 4 (second row), a region with low initial attention becomes more prominent in later stages, demonstrating the adaptiveness of our selection process. (See also Reviewer Ekip W6.)
>
> **W4: How were hyperparameters determined in the experiments?**
>
> Thank you for raising this important question.
> Regarding $\lambda$, while each setting has its optimal value Tab. 6, we found that $\lambda = 1.0$ consistently performs well across most conditions. This supports our recommendation of $\lambda = 1.0$ as a robust default, further backed by the trend in Fig. 6(a), where performance peaks around this value.
> As for $\alpha$, $\beta$, and $\gamma$, we acknowledge the difficulty of manual tuning. Inspired by recent works [1], which explore adaptive parameter selection, we applied divergence based adaptive parameter selection. This dynamic adjustment allowed SECOND to perform close to manually-tuned baselines, achieving reasonable results on the benchmark POPE.
> We believe this reduces hyperparameter sensitivity and improves the practicality of SECOND, and we plan to further develop this adaptive approach in future work.
>
> \\(\Delta logit_{s} = α * (logit_{s} - logit_{s-1}),\\)
>
> where \\(α = 1 - D_{bd}(S||S-1),\\)
>
> and $D_{bd}(S||S-1)$ denotes the bounded divergence of token distribution between stage $s$ and $s-1$. (from [1])
>
> |  | POPE |
> | --- | :---: |
> | LLaVA-NeXT Vicuna 7B | 86.5 |
> | LLaVA-NeXT Vicuna 7B + SECOND w/ $D_{bd}$ CD | 88.2 |
> | LLaVA-NeXT Vicuna 7B + SECOND w/ Hard CD | **89.2** |
>
> [1] "Code: Contrasting self-generated description to combat hallucination in large multi-modal models." NeurIPS 2024
>
> **W5: While experiments do show benefits of SECOND across different setups, there are still some cases where performance gains are marginal or even negative.**
>
> Thank you for pointing out this important issue. While SECOND does not yield improvements uniformly across all models and benchmarks, it is important to highlight that its primary goal is to mitigate object hallucination in VLMs. In this regard, it shows strong effectiveness—outperforming baselines in 11 out of 12 settings on the POPE benchmark (as shown in Tab. 1), which directly targets perceptual hallucination.
> It also consistently improves performance on object-centric tasks such as VQAv2 across all tested models, highlighting its benefit for fine-grained object understanding. Minor drops observed in general-purpose benchmarks (e.g., MMStar, MMBench) may result from untrained patch selection slightly disrupting attention patterns—an area we plan to explore further.
> Despite a few exceptions, the overall trend shows that SECOND offers reliable gains on object-level tasks. We have revised the manuscript to clarify both the strengths and limitations of our approach.
>
> **W6: How does computational complexity of SECOND compare to that of VCD?**
>
> Please refer to our response to Reviewer KN5t W2, where we provide a detailed comparison of per-token generation time across these methods.
>
> **W7: Presentation of the method could be improved with more details about the method**
>
> With responses to W2 and W3, we have revised and extended the explanations in Sec. 3 and Sec. 4 to improve the clarity and readability of our method.
>
> **W8: nit: L not defined in def 3.1**
>
> Thank you for pointing this out. In Def. 3.1, L refers to the length of the generated sequence. We have clarified this in the revised manuscript to avoid confusion.

---

> > ### Comment · Reviewer_i6dn · 2025-04-03
> >
> > I thank the authors for the thorough reply, I raised my score accordingly.

---

### Official Review · Reviewer_Ekip · 2025-03-11

**Overall Recommendation:** 3

**Summary:**

This paper focuses on how to improve the hallucination of MLLMs. Firstly, the paper conducts some analysis of hallucinations in MLLMs, proposing two metrics, namely the Hallucination Probability and the Attention Dice Coefficient, and introducing the research motivation of needing to enhance the model's visual perception of objects. Subsequently, this paper presents a training-free method to alleviate the issues of MLLMs, named SECOND (Selective and Contrastive Decoding). This method first utilizes the model's self-attention to perform inference serially on multi-scale images, screening out the patches of the main objects of interest for the next stage of inference. Then, it fuses the inference logits of multiple stages and decodes them through contrastive decoding to obtain the final result. Finally, comparative experiments are carried out on the proposed method modules using three models, namely LLaVA-Next, LLaVA-OneVision, and Yi-VL. Additionally, ablation experiments are conducted on hyperparameters in multiple models to demonstrate the effectiveness of SECOND.

**Claims And Evidence:**

yes

**Essential References Not Discussed:**

No

**Experimental Designs Or Analyses:**

Yes

**Methods And Evaluation Criteria:**

yes

**Other Comments Or Suggestions:**

1. The details of the method section are unclear：a) How is the visual attention used to filter patches calculated in the method, and to whom is the attention calculated? b) Is the visual path input in the current stage all from the previous stage or only from the current stage

**Other Strengths And Weaknesses:**

Strengths:
1. This paper provides a certain degree of analysis on the hallucination of MLLM, which may inspire some new research ideas.
2. The method proposed in this paper is training-free, capable of conducting experiments on pre-trained models, and has low resource consumption.
Weaknesses：
1. It is very unreasonable in the theoretical analysis section of the paper: firstly, the formula definition of Hallucination probability proposed in Section 3 seems to lack support from relevant theoretical papers (not mentioned in the paper); Next, only use Fig 2 (a) It cannot be proven that 'Non aggregated responses preferentially exhibit lower hallucination probabilities'. Fig. 2 (a) shows two edge distributions, which can only prove that if it is known to be a Non exhaustive response, it has a higher probability of being a lower hallucination probability. Because when it is hallucinated responses, the values of these hallucination probabilities are nearly uniformly distributed.
2. Due to the hallucination of MLLM not only existing in object-centric questions or tasks, but also in more general tasks, the experimental part lacks validation of more powerful models and common benchmarks, otherwise it is difficult to prove the generality of its method. Even if it cannot be validated, the author needs to explain the specific reasons: a) the model part includes more advanced qwen2-vl and intervl2 b) the benchmark part includes MMUL/MMMU pro/MegaBenchmark, etc

**Questions For Authors:**

1. The problem of cumulative error in SECOND visual attention: SECOND uses multi-stage visual attention for patch selection, where the attention calculated in the current stage is used for the next stage of patch selection.This method seems unable to correct the already erroneous visual attention and introduces cumulative errors. For example, in the first stage, there is an error in the attention, that is, when the attention is completely focused on visual areas unrelated to the prompt, then the input patch in the next stage will have errors and cannot be corrected. So, how to improve this problem?

**Relation To Broader Scientific Literature:**

I think this work is tradition topic with a somewhat novel method.

**Theoretical Claims:**

Yes

---

> ### Author Rebuttal · Authors · 2025-04-01
>
> Dear Reviewer Ekip,
>
> We really appreciate your thorough review of our paper. We address the raised concerns and questions below.
>
> **W1: The formula definition of Hallucination probability proposed in Sec. 3 seems to lack support from relevant theoretical papers (not mentioned in the paper)**
>
> We sincerely appreciate your feedback. The definition of Hallucination Probability in Sec. 3 builds on prior works [1, 2], where the probability of a valid sequence is defined as the product of token-level probabilities:
>
> \\(p(y|x,c) = \prod^T_{t=1} p(y_t|y_{<t}, x, c), \quad \text{where } y_{<t} \triangleq [y_0, ..., y_{t−1}]. \\)
>
> We define Hallucination Probability as its complement, $1 - p(y|x,c)$, representing the likelihood of generating a hallucinated response. We have revised the manuscript to include these references and clarify the motivation.
>
> [1] "Multi-modal hallucination control by visual information grounding.", CVPR. 2024.
>
> [2] "Looking for a needle in a haystack: A comprehensive study of hallucinations in neural machine translation." ACL. 2023.
>
> **W2: Only use Fig. 2 (a) It cannot be proven that 'Non aggregated responses preferentially exhibit lower hallucination probabilities'.**
>
> Thank you for raising this important point. As you noted, Fig. 2(a) shows that $P_{\text{Hal}}$ is typically lower for non-hallucinated responses, while hallucinated ones exhibit a more uniform distribution. We agree that this does not imply hallucinated responses always have higher $P_{\text{Hal}}$.
> However, the converse holds empirically: responses with higher $P_{\text{Hal}}$ are more likely to be hallucinated. For instance, in the [0.4, 0.5) range, the valid-to-hallucinated ratio is approximately 1:9, indicating a strong correlation. Additionally, Fig. 2(b) shows that higher $P_{\text{Hal}}$ aligns with lower Attention Dice scores, suggesting weaker visual grounding.
> Our aim was to show that such high-risk responses can be mitigated by SECOND, which both reduces $P_{\text{Hal}}$ and improves grounding. We will revise the manuscript to clarify this interpretation and better explain the rationale.
>
> **W3: a) the model part includes more advanced qwen2-vl and intervl2 b) the benchmark part includes MMUL/MMMU pro/MegaBenchmark, etc**
>
> Thank you for your insightful comment. Due to the limited time during the rebuttal period, implementing SECOND on newly released models posed some challenges. Nevertheless, we have conducted additional experiments to compare SECOND with a broader range of more advanced models.
> |  | POPE (pop, f1) |
> | --- | :---: |
> | Qwen2-VL 7B | 87.9 |
> | InternVL2 4B | 87.3 |
> | InternVL2 8B | 86.7 |
> | InternVL2 26B | 87.8 |
> | Ivy VL 3B | 87.5 |
> | Ovis 8B | 88.6 |
> | LLaVA-NeXT Vicuna 7B + SECOND | **89.2** |
> | LLaVA-NeXT Mistral 7B + SECOND | **88.8** |
>
> As shown above, SECOND outperforms advanced models on the object hallucination task.
> Regarding the benchmarks you mentioned, we would like to clarify that MMLU is primarily designed to evaluate LLM capabilities. For MegaBench, the length of the prompts is extremely large, and in some cases exceeds the max_token_length of models such as LLaVA-NeXT Vicuna. Therefore, we conducted additional experiments on MMMU-Pro instead. We kindly refer you to Reviewer KN5t W1 for further details.
>
> **W4: How is the visual attention used to filter patches calculated in the method, and to whom is the attention calculated**
>
> Thank you. We revised the manuscript for clarity. For full details, please see our response to Reviewer Lfoe W3.
>
> **W5: Are patches from previous stages reused?**
>
> Yes, visual patches are accumulated across stages. Each stage uses both newly selected and retained patches to refine grounding. While this was indicated in Fig. 1, Fig. 3 and Sec. 4.1, we have now clarified it further.
>
> **W6: seems unable to correct the already erroneous visual attention and introduces cumulative errors.**
>
> Thank you for this insightful observation. We clarify that patch selection in SECOND is not restricted to a fixed or narrowing set of regions. As described in Sec. 4.1, the selection ratio ($P_{\text{select}}$) is dynamically adjusted across stages based on changes in attention entropy. When entropy increases—signaling higher uncertainty—the selection ratio can also increase, allowing inclusion of previously unselected but relevant patches.
> Our Contrastive Decoding further ensures that newly added patches, even if missed earlier, can still influence the final output—helping correct early-stage attention errors. As shown in Fig. 4, some patches initially receive high attention but decrease as new patches are added. Also in the second row of Fig. 4, we observe the model gradually shifting its focus toward the top-right region, which initially had low attention.
> To address cases of completely misaligned attention, we compare SECOND with a variant using all patches (no selection) in Tab. 4. SECOND outperforms this baseline, showing its robustness against cumulative attention errors.

---

### Official Review · Reviewer_Lfoe · 2025-03-12

**Overall Recommendation:** 4

**Summary:**

In this paper, a decoding method for LVLMs named SECOND is proposed. SECOND consists of selective multi-scale feature integration and multi-stage contrastive decoding. The first method, selective multi-scale feature integration leverages multi-scale feature map with patch selection scheme, where important patches are progressively selected in high-resolution features based on lower resolution attention values. Multi-stage contrastive decoding contrasts outputs obtained with multi-scale features, thereby prioritizing outputs of relative ‘experts’ with better feature maps.

**Claims And Evidence:**

The authors’ claims are supported with proper observations and analyses.

**Essential References Not Discussed:**

Since there are multiple works [1-6] utilizing contrastive decoding to mitigate hallucination problem of LVLMs, discussion with those works should be provided.

[1] Huo et al., Self-Introspective Decoding: Alleviating Hallucinations for Large Vision-Language Models, ICLR 2025

[2] Zhuang et al., VASparse: Towards Efficient Visual Hallucination Mitigation for Large Vision-Language Model via Visual-Aware Sparsification, CVPR 2025

[3] Cho et al., Do You Keep an Eye on What I Ask? Mitigating Multimodal Hallucination via Attention-Guided Ensemble Decoding, ICLR 2025

[4] Kim et al., VACoDe: Visual Augmented Contrastive Decoding, ICML Workshop 2024

[5] Park et al., ConVis: Contrastive Decoding with Hallucination Visualization for Mitigating Hallucinations in Multimodal Large Language Models, arXiv 2024

[6] Lee et al., Delve into Visual Contrastive Decoding for Hallucination Mitigation of Large Vision-Language Models, arXiv 2024

**Experimental Designs Or Analyses:**

Proper ablation studies are provided, which validates the effectiveness of each method.

**Methods And Evaluation Criteria:**

The proposed methods are sound. Also, experiments on POPE benchmark and various multimodal benchmarks supports the method.

**Other Comments Or Suggestions:**

Although the paper is generally well-written, there are some areas that could be improved:

1. L153 (left column): The implication of the attention Dice coefficient could be briefly introduced for better clarity. For example, consider adding a sentence like: *“A higher attention Dice coefficient indicates that an LVLM properly focuses on objects.”*
2. L163 (left column): The paper states that *“hallucinated responses tend to have higher Attention Dice scores.”* However, in Figure 2-(b), hallucinated responses appear to have **lower** scores compared to non-hallucinated responses. Is this a mistake in the paper, or have I misunderstood?
3. L255 (left column): The explanation of how visual attention is generated and interpreted is not detailed enough. Additionally, the selection process is overly simplified. While Section D of the supplementary material provides further details, it would be better to introduce more information in the main paper and explicitly reference the exact section of the supplementary material.

**Other Strengths And Weaknesses:**

N/A

**Questions For Authors:**

Please kindly address concerns in other sections.

**Relation To Broader Scientific Literature:**

This work provides a simple yet effective method to enhance LVLMs by effectively utilizing multi-scale features. Also, the observations provided in the paper help understanding problems of LVLMs.

**Theoretical Claims:**

The authors make some theoretical claims that are supported with observations and proofs.

---

> ### Author Rebuttal · Authors · 2025-04-01
>
> Dear Reviewer Lfoe,
>
> We greatly appreciate your valuable feedback on our paper. We address the raised concerns and questions below.
>
> **W1: L153 (left column): The implication of the attention Dice coefficient could be briefly introduced for better clarity.**
>
> Thank you for the insightful comment. To improve clarity, we have revised main manuscript by adding a brief explanatory sentence, “A higher Attention Dice coefficient indicates that an LVLM properly focuses on objects”. This addition will help readers better understand the meaning and importance of the Attention Dice Coefficient in our theorem.
>
> **W2: The paper states that “hallucinated responses tend to have higher Attention Dice scores.” However, in Fig. 2 (b), hallucinated responses appear to have lower scores compared to non-hallucinated responses. Is this a mistake in the paper, or have I misunderstood?**
>
> Thank you for pointing this out. You are correct — there is a mistake in the original text. Hallucinated responses actually tend to have lower Attention Dice Coefficient, as correctly shown in Fig. 2 (b). We have corrected the sentence to reflect this and apologize for the confusion.
>
> **W3: L255 (left column): The explanation of how visual attention is generated and interpreted is not detailed enough. Additionally, the selection process is overly simplified. While Section D of the supplementary material provides further details, it would be better to introduce more information in the main paper and explicitly reference the exact section of the supplementary material.**
>
> We are grateful for your constructive suggestion. To clarify, we describe the computation of attention in Section 3.2 (for calculating the Attention Dice Coefficient), and we adopt the same formulation in our methodology (Sec. 4).
> The visual attention used to filter patches is derived from two sources:
> 1.	Self-attention maps from the vision encoder of the LVLM, which operate over 2D spatial image patches.
> 2.	Cross-attention maps from the LLM decoder, which capture the attention from generated textual tokens to the visual tokens (i.e., image patches).
> To compute a unified attention signal that reflects both visual and textual modalities, we multiply these two attention maps element-wise. Since the cross-attention is defined over flattened visual tokens (i.e., in a 1D format), we first map the 1D token-based attention values back to their corresponding 2D spatial coordinates to align with the vision encoder’s attention layout. This alignment enables meaningful element-wise fusion of the two attention maps.
> This fused attention allows us to effectively estimate the contribution of each image patch to the generated tokens, which we then use to guide patch filtering in our method. We will revise the manuscript to make this computation process more explicit and improve overall clarity.
>
> **W4: Essential References Not Discussed**
>
> We sincerely thank the reviewer for highlighting this important references. We fully agree that recent decoding-based approaches—such as VASparse, VACoDe, ConVis, SID, and Ensemble Decoding (ED)—have made important contributions to hallucination mitigation in vision-language models (VLMs). To provide a more comprehensive and contextualized discussion, we have incorporated these references into the Sec. 2. Related Works of the revised manuscript. Furthermore, we have added explicit comparisons between each of these methods and our proposed SECOND approach in the main text, highlighting key distinctions in terms of decoding strategy, use of contrastive signals, reliance on external augmentations, and computational characteristics. We hope this addition strengthens the clarity of our contributions and the positioning of our work within the existing literature.

---

> > ### Comment · Reviewer_Lfoe · 2025-04-02
> >
> > I have carefully read the authors' rebuttal as well as their responses to other reviewers.
> > Overall, I find the paper valuable, and my concerns regarding its clarity have been adequately addressed.
> > Therefore, I maintain my original rating.

---

### Official Review · Reviewer_KN5t · 2025-03-13

**Overall Recommendation:** 3

**Summary:**

This paper introduces SECOND, a training-free approach to mitigate perceptual hallucination in LVLM.  More specifically, it progressively refines (by patch selection) multi-scale visual information in an object-centric manner, and uses multi-stage contrastive decoding to reduce perceptual hallucinations.  Results show it outperforms baselines across diverse benchmarks.

## update after rebuttal

The rebuttal has clarified my concerns. I am happy to maintain my original recommendation.

**Claims And Evidence:**

The experiments, evaluations, analysis and theory proof support its claim.

**Essential References Not Discussed:**

no

**Experimental Designs Or Analyses:**

The authors implement SECOND on three recent LVLMs: LLaVA-NeXT, LLaVA-OneVision, and Yi-VL; evaluate these models on the POPE benchmark for perceptual hallucination and VQAv2, MMStar, and MMBench for general tasks. Qualitative results are also presented to give readers insight.

**Methods And Evaluation Criteria:**

The methods involve two core components: 1) adaptive multi-scale patch selection, 2) multi-stage contrastive decoding. Ablations show both are important in mitigating perceptual hallucinations.

The authors use the POPE benchmark to evaluate perceptual hallucination, and use VQAv2, MMStar, and MMBench for general tasks. However, more tasks could be introduced to verify SECOND’s effectiveness on common VLM tasks, such as captioning, document understanding, infographics reasoning etc.

**Other Comments Or Suggestions:**

no

**Other Strengths And Weaknesses:**

Strengths
- The ideal is novel and further strengthened by the theory proof.
- The paper is well written and easy to read.
- Theories are presented clearly.

Weaknesses
- It’s unclear how efficient the method is, with additional multi-stage computation.
- As mentioned above, more common VLM tasks can be added to verify the effectiveness of the methods, such as in document instead of semantic understanding.

**Questions For Authors:**

no

**Relation To Broader Scientific Literature:**

This paper successfully extends the concept of Contrastive Decoding from LLMs to LVLMs, which might be insightful for the community.

The theories and proofs presented in this paper also provide valuable insights for the community.

**Theoretical Claims:**

The authors raise the hypothesis that a model’s ability to focus accurately on target objects significantly reduces hallucination risk, which was proved via Attention Dice Coefficient experiments. Then it proves that multi-stage patch selection increases the Attention Dice Coefficient, thereby reducing the probability of hallucination.

---

> ### Author Rebuttal · Authors · 2025-04-01
>
> Dear Reviewer KN5t,
>
> Thanks for your valuable feedback! We provide point-by-point responses to address your concerns below.
>
> **W1: More tasks could be introduced to verify SECOND’s effectiveness on common VLM tasks, such as captioning, document understanding, infographics reasoning etc.**
>
> Thank you for this valuable suggestion. To further substantiate the effectiveness of SECOND on a broader range of common VLM tasks, we have conducted additional experiments on MMMU_Pro, complementing the previously reported results on MMStar and MMBench.
> These benchmarks collectively provide a comprehensive evaluation across a diverse set of task categories, including instance relation reasoning, diagram reasoning, calculation, and scene understanding—covering many of the task types you mentioned. The new result of MMMU Pro, presented below, offer additional evidence supporting the robustness and generalizability of SECOND across diverse VLM challenges.
> This additional experiment has been added to Appendix of the main manuscript. We hope this additional evaluation addresses your concern.
>
>
> |  | CD | MMMU_Pro |
> | --- | :---: | :---: |
> | LLaVA-NeXT Vicuna 7B | X | 16.1 |
> | LLaVA-NeXT Vicuna 7B + SECOND | X | 16.7 |
> | LLaVA-NeXT Vicuna 7B + SECOND | O | 16.9 |
> | LLaVA-OneVision Qwen 0.5B | X | 14.5 |
> | LLaVA-OneVision Qwen 0.5B + SECOND | X | 15.0 |
> | LLaVA-OneVision Qwen 0.5B + SECOND | O | 15.0 |
>
> **W2: It’s unclear how efficient the method is, with additional multi-stage computation**
>
> Thank you for highlighting this important point. To address concerns regarding the computational efficiency of SECOND, we have measured and compared the per-token generation time of our method against the baseline VCD and the method OPERA[1] (a widely used approach for mitigating VLM hallucination, proposed by Reviewer i6dn).
> The results are summarized as follows:
>
> |  | VCD | SECOND(3-stages) | SECOND(4-stages) | OPERA |
> | --- | :---: | :---: | :---: | :---: |
> | LLaVA-NeXT Vicuna 7B (sec / token) | 0.41 | 1.28 | 1.80 | 5.00 |
> | Yi-VL 6B (sec / token) | 0.26 | 0.73 | - | 4.79 |
>
> These measurements demonstrate that while SECOND introduces a modest increase in computation due to its multi-stage patch selection, the increase remains manageable. Importantly, this additional computational cost is well-justified by the consistent performance improvements on hallucination mitigation and general VLM performance across our experiments. We will include these results in the revised manuscript to provide a clearer picture of SECOND’s efficiency.
>
> [1] "Opera: Alleviating hallucination in multi-modal large language models via over-trust penalty and retrospection-allocation." CVPR. 2024.

---

### Decision · Program_Chairs · 2025-05-01

**Decision:**

Accept (poster)

**Comment:**

After the rebuttal, all reviewers have raised their rating to positive (1 Accept and 3 Weak Accept). The previous main concerns are about the presentation clarity. After the rebuttal, the authors have addressed these concerns. Thus, we recommend **Accept** for this submission. It is necessary to incorporate all these rebuttal discussions into the camera ready version.